# Progress and Perspective of Glass-Ceramic Solid-State Electrolytes for Lithium Batteries

**DOI:** 10.3390/ma16072655

**Published:** 2023-03-27

**Authors:** Liyang Lin, Wei Guo, Mengjun Li, Juan Qing, Chuang Cai, Ping Yi, Qibo Deng, Wei Chen

**Affiliations:** 1School of Aeronautics, Chongqing Jiaotong University, Chongqing 400074, China; 2Chongqing Key Laboratory of Green Aviation Energy and Power, Chongqing 401130, China; 3The Green Aerotechnics Research Institute, Chongqing Jiaotong University, Chongqing 401120, China; 4School of Materials Science and Engineering, Chongqing Jiaotong University, Chongqing 400074, China; 5Key Laboratory of Hebei Province on Scale-Span Intelligent Equipment Technology, Tianjin Key Laboratory of Power Transmission and Safety Technology for New Energy Vehicles, School of Mechanical Engineering, Hebei University of Technology, Tianjin 300401, China

**Keywords:** lithium batteries, glass-ceramic, solid electrolyte, synthesis and characterization, high ionic conductivity

## Abstract

The all-solid-state lithium battery (ASSLIB) is one of the key points of future lithium battery technology development. Because solid-state electrolytes (SSEs) have higher safety performance than liquid electrolytes, and they can promote the application of Li-metal anodes to endow batteries with higher energy density. Glass-ceramic SSEs with excellent ionic conductivity and mechanical strength are one of the main focuses of SSE research. In this review paper, we discuss recent advances in the synthesis and characterization of glass-ceramic SSEs. Additionally, some discussions on the interface problems commonly found in glass-ceramic SSEs and their solutions are provided. At the end of this review, some drawbacks of glass-ceramic SSEs are summarized, and future development directions are prospected. We hope that this review paper can help the development of glass-ceramic solid-state electrolytes.

## 1. Introduction

Since Sony first commercialized lithium-ion batteries (LIBs) in 1991, LIBs have been widely used in electronics, power and energy storage applications due to their high working voltage, high energy density, long cycle life and no memory characteristics [1,2,3]. With the rapid development of electric vehicles (EVs), traditional LIBs have been insufficient to meet the range of EVs. The energy density of traditional LIBs has achieved 260 Wh·kg^−1^, which is approaching the limitations of traditional LIBs [4]. Metal lithium has a high theoretical specific capacity (3860 mAh·g^−1^) and the lowest redox potential (−3.04 V vs. SHE) and can effectively increase the energy density of the battery when used as the anode [5]. However, traditional liquid electrolytes restrict the application of the lithium-metal anode because they contain flammable organic solvents that cause some safety problems [6,7]. All-solid-state lithium-metal batteries (ASSLMBs) with higher safety and higher energy density composed of lithium-metal anodes and solid-state electrolytes (SSEs) instead of traditional liquid electrolytes are expected to become the next generation of lithium battery.

In 1833, Faraday first discovered the ionic conductivity of solid Ag_2_S and PbF_2_, and research on the ionic conductivity of solids has been conducted since that time [8]. In the 1960s, Na_2_O·11Al_2_O_3_ with Na^+^ ion conductivity was discovered, and researchers discovered that this type of material possessed the property of high ionic conductivity and had the potential to be used as SSEs [9]. Therefore, using solids with satisfactory ionic conductivity to form ASSLIBs became possible. SSEs, the most important component of ASSLIBs, have many advantages over liquid electrolytes.

The non-flammable characteristics of SSEs make ASSLIBs have higher safety performance than LIBs [10].Compared to traditional LIBs, SSEs are able to replace the liquid electrolyte and separator to effectively reduce battery weight. Meanwhile, the energy density of the battery is increased by combining the application of a lithium-metal anode [11].Compared to conventional LIBs, ASSLIBs have greater structural design advantages because they can be connected in series internally to achieve higher voltages. Chen et al. [12] stacked one, two and three solid-state cells in a button battery to obtain open-circuit voltages of 3.08, 6.51 and 9.12 V, respectively.

Although ASSLIBs have certain advantages, their process of industrialization is still limited by technological, marketing and financial factors. On the technological side, the research of SSE synthesis method, stability, conductivity and interfacial properties is the key to practical application. After years of development, SSEs can be divided into three categories: inorganic solid electrolytes (ISEs), polymer solid electrolytes (PSEs) and composite solid electrolytes (CSEs). Among them, ISEs can be divided into amorphous glass, glass-ceramic and polycrystalline ceramic. Glass is an amorphous supercooled liquid, while glass-ceramics are partially crystalline glasses, consisting of a mixture of crystalline and amorphous glass phases [13,14]. The definition of glass-ceramic materials is an inorganic non-metal material prepared by controlling the crystallization of glass through different processing methods [15]. They consist of at least one functional crystalline phase and residual glass. The volume fraction of the crystalline part in glass-ceramic materials is typically in the range of 10–90% [14]. The main advantages of glass-ceramic materials are their dense, non-porous microstructure, and good mechanical, electrical and thermal properties. Glass-ceramic SSEs have become one of the hot research directions for SSEs due to their excellent ionic conductivity, electrochemical properties and better compatibility with electrodes.

Glass-ceramic SSEs are divided into two main categories, oxide glass-ceramic SSE systems and sulfide glass-ceramic SSE systems. Oxide glass-ceramic SSEs include NASICON-type electrolytes and some other oxides. They are mainly prepared by the melt-quenching method with subsequent heat treatment, and their main advantages are high ionic conductivity (10^−4^~10^−3^ S·cm^−1^), large Li^+^ transference number and high mechanical strength [16,17]. The sulfide glass-ceramic SSEs are mainly Li_2_S-P_2_S_5_ binary systems, which are prepared by mechanical ball milling and subsequent heat treatment, and their main advantages are high ionic conductivity (10^−3^~10^−2^ S·cm^−1^) [18,19,20,21]. Although glass-ceramic SSEs generally have high ionic conductivity, the stability of the SSE itself and the interface problems between the electrode/electrolyte are major impediments to the practical application of ASSLIBs [22,23]. Improving the properties including ionic conductivity and chemical stability has become one of the main focuses of current research on glass-ceramic SSEs.

In this review, first, the synthesis and characterization of glass-ceramic electrolytes in recent years will be summarized. At the same time, the ionic conduction mechanism and the high ionic conductivity of glass-ceramic SSEs will be introduced briefly in this work. Then, we will discuss the common interface problem between SSEs and electrodes and summarize the performance of glass-ceramic SSEs and the corresponding solutions to the interface problem. We hope to provide reference for the development of the ASSLIB industry by reviewing the research progress of glass-ceramic SSEs and looking forward to their application prospects.

## 2. Ionic Conduction Mechanism

For designing high-performance SSEs, an understanding of their ion conduction mechanisms is necessary. Li^+^ ion migration in ceramics relies on different types of defects, including point defects, line defects, planar defects, volume defects and electron defects. Compared to other defects, point defects have a greater impact on cation transport in crystals [24]. In a perfectly ordered crystal, ions cannot leave their host position [8]. The migration of ions in SSEs is accomplished by moving point defects in the crystal.

The basic assumption about the ionic conduction mechanism in polycrystalline (ceramic) is that vacancies in the lattice and interstitial spaces in the cationic sublattice are considered as charged movable species [25,26]. It is noteworthy that only a fraction of cations in a lattice has an ability to move having vacant stable or meta-stable lattice nodes within reach [9]. Currently, there are three main types of cation migration, as shown in Figure 1.

Cation vacancy diffusion, cation migration from the initial position to its adjacent vacancy lattice position.The cation occupying the interstitial migrates directly to the adjacent vacant interstitial.Interstitialcy mechanism, cation occupying a lattice interstitial migrates to an adjacent lattice node, migrating the cation occupying that lattice to the next site.

For polycrystalline ceramic SSEs, the Li^+^ transport mechanism depends on three factors: carrier type, diffusion pathways and diffusion type. The carrier type and concentration are determined by the point defects in the polycrystalline ceramic structure, which directly affect the ionic conductivity. The interactions between Li ions during migration in the crystal and between ones and the surrounding environment will significantly affect the ionic conductivity [24,27,28,29].

**Figure 1 materials-16-02655-f001:**
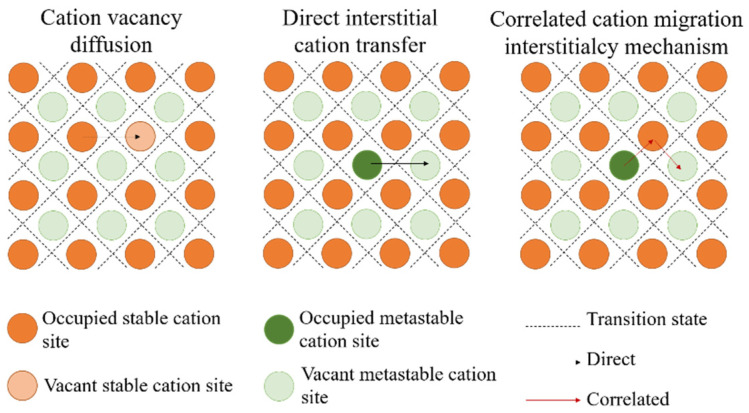
Three typical cation migration mechanisms: cation vacancy diffusion, direct interstitial cation transfer, correlated cation migration interstitialcy mechanism. Reprinted with permission from ref. [26]. Copyright 2019 Springer Nature.

Compared to ceramic SSEs, amorphous (glass) SSEs have better flexibility, uniformity and density. Meanwhile, the glass SSEs show no grain boundary resistance and isotropic Li^+^ mobility. These properties of glass SSEs have prompted attempts to find its ionic conduction mechanism. At present, although many experimental data on Li^+^ conduction in glass SSEs are available, the Li^+^ conduction mechanism in glass SSEs is still not well explained, and no relevant general theory has been established. The main challenge is that glass SSEs are a short-range ordered, long-range disordered amorphous material. It means that the glass SSEs have no long-range crystalline order, no regular symmetric long-range ion migration pathways and no regular symmetric short-range coordination order [9]. In glass SSEs with disordered structure, the migration of cations in SSEs cannot be explained by a single factor. During the migration of cations in glass SSEs, charge carrier interactions and even interactions with the transport matrix can have an effect on the migration of ions. This makes the theoretical development of the conduction mechanism of cations in glass SSEs difficult. However, hypotheses have been offered to explain how the cations migrate in the amorphous SSE [30].

Funke et al. [31] suggested that structure and kinetic disorder are major factors in the high ionic conductivity of amorphous materials. They defined the movement of ions in a completely ordered crystal structure as level 1. This material is regarded as an insulator without ion movement because of the absence of defects in the perfectly ordered crystal structure. Crystal structures with few defects are defined as level 2, and a single point defect can only move randomly to another location. Materials with disordered structure are defined as level 3, and ion movement cannot be described by defect theory but is related to multiple interactions with the surrounding environment. They suggested that the mismatch caused by the hopping of ions resulted in rearrangement of the particles of the neighborhood. The hopping ion is accommodated by the new site created by its neighborhood relaxation.

## 3. Synthesis and Characterization of Glass-Ceramic Solid-State Electrolytes

Currently, there are two main types of glass-ceramic SSEs, oxide glass-ceramic SSE systems and sulfide glass-ceramic SSE systems. Glass-ceramic SSEs are mainly prepared in two steps by the melt-quenching method and mechanical ball-milling method. In the first step, the required parent glass is prepared at a given ratio of raw materials by high temperature melting or mechanical ball milling. In the second step, the parent glass is heat-treated between the glass transition temperature T_g_ and the crystallization temperature T_c_. T_g_ and T_c_ are determined by differential thermal analysis (DTA) and differential scanning calorimetry (DSC). Currently, glass-ceramic materials with high ionic conductivity are mainly obtained by changing the optimized raw material ratio, heat treatment temperature and time. Glass-ceramic SSEs prepared by wet chemical methods have also been reported recently [32,33,34,35]. For the prepared glass-ceramic SSEs, the properties were investigated mainly by characterization means such as impedance spectroscopy (IS), X-ray diffraction (XRD), DTA, DSC and electron microscopy. In some cases, short-range ordering in glass-ceramic SSEs has also been investigated by nuclear magnetic resonance (NMR). In this chapter, the preparation and characterization of oxide glass-ceramic SSE systems and sulfide glass-ceramic SSE systems are highlighted in the following sections. Additionally, possible ways to improve their ionic conductivity will be discussed.

### 3.1. Oxide Glass-Ceramic SSE Systems

Most of the oxide SSEs are polycrystalline ceramic SSEs whose advantages are high ionic conductivity, high mechanical strength and a wide electrochemical stability window. However, the interface problem between this type of SSEs and electrodes is more prominent. Compared to polycrystalline ceramics, glass has certain advantages in terms of flexibility, homogeneity and density. Therefore, glass-ceramic SSEs are prepared by fusion glass and partial crystallization of glass, which not only improve the ionic conductivity but also optimize the interface between SSEs and electrodes to some extent. The current research on oxide glass-ceramic SSE systems is mainly focused on Na^+^ superionic conductor (NASICON)-type SSEs and some other types of oxides.

#### 3.1.1. NASICON-Type Glass-Ceramic Systems

In 1976, the NASICON-type compound was first discovered by Goodenough et al. [36]. The chemical formula is NaM_2_(PO_4_)_3_ (M is a tetravalent metal [M^4+^], e.g., Ge, Ti, Sn and Zr [37]). Na_1+x_Zr_2_Si_x_P_3−x_O_12_ (0 < x < 3) which is called NASICON and is obtained when the P is partially replaced by Si. Their structures have a rhombic crystal lattice, space group R-3c, but for some compounds the trigonal distortion of the lattice was found [22,38]. Figure 2 shows a typical crystal structure of this type of compound, which consists of stacked (or joint) MO_6_ octahedra and PO_4_ tetrahedra [39]. The charge carriers in the structure can occupy two different six-coordinated positions, M1 between two MO_6_ octahedra and M2 in the eight-coordination position between two rows of MO_6_ octahedra. Li^+^ migrates in the ion channel formed by M1 and M2 under the influence of the electric field, and all positions of the occupied part form the 3D channel of Li^+^ [40,41]. Li^+^ conduction can be achieved by replacing Na with Li while maintaining the crystal structure, and the most representative one is LiTi_2_(PO_4_)_3_ [42]. Lithium analogues of NASICON-type compounds are heavily investigated as promising SSEs for ASSLIBs.

In recent years, there are mainly two types of NASICON-type SSEs, LATP and LAGP. The representative materials for LATP and LAGP are Li_1.3_Al_0.3_Ti_1.7_(PO_4_)_3_ [43] and Li_1.5_Al_0.5_Ge_1.5_(PO_4_)_3_ [44], respectively. The ionic conductivity of SSEs is mainly affected by the preparation process, microstructure and porosity. Due to the open framework structure of NASICON, this type of SSE generally suffers from high porosity and high grain boundary resistance, which leads to the low total conductivity of SSEs [16,45]. The low void fraction of glass-ceramic materials can improve their cation migration properties. At the same time, glass-ceramic materials have better conductive interface regions on newly formed crystalline grains embedded in the glass matrix, and the grain boundary resistance can be effectively reduced by controlling the crystallization of the parent glass. Therefore, the electrical properties of NASICON-type glass-ceramic SSEs can be well improved as a result of optimizing the synthesis conditions.

In most studies, scholars have mainly used the melt-quenching method [46,47,48,49] to prepare NASICON-type glass-ceramic SSEs, which is divided into three main steps: (1) the mixture of raw materials is melted at high temperatures to form precursors, (2) rapid cooling to form the parent glass and (3) after annealing to release stress, the glass undergoes a period of heat treatment to nucleate and grow NASICON crystals. The control and optimization of various parameters are very important for the preparation of glass-ceramic SSEs by the melt-quenching method, such as the composition ratio of elements, and the temperature of crystallization and annealing [50]. An improper elemental composition ratio can lead to the formation of impurity phases in NASICON glass-ceramic SSEs, which can hinder the migration of Li^+^ ions leading to a decrease in ionic conductivity. In contrast, a proper crystallization temperature can result in glass-ceramic SSEs with low void fraction and grain boundary resistance. Illbeigi et al. [51] synthesized Li_1+x+y_Al_x_Cr_y_Ge_2−x−y_(PO_4_)_3_ by melt quenching (x + y = 0.5, y = 0, 0.1 0.25, 0.4, 0.5 and x = 0.5, 0.4, 0.25, 0.1, 0) glass-ceramic SSEs. It was found that the addition of Cr can increase the crystal cell dimension, thus increasing their electrical conductivity. The prepared Li_1.5_Al_0.4_Cr_0.1_Ge_1.5_(PO_4_) glass-ceramics not only have a high ionic conductivity but also show an excellent electrochemical stability window up to 7 V vs. Li/Li^+^. However, when the content of Cr > 0.1, the authors found the impure phases GeO_2_ and CrPO_4_ in the grain boundaries by XRD and FESEM. Additionally, the impure phase hinders the migration of Li^+^ ions and causes a decrease in the ionic conductivity of the materials; the XRD patterns are shown in Figure 3a. The maximum Li^+^ conductivity of Li_1.5_Al_0.4_Cr_0.1_Ge_1.5_(PO_4_)_3_ sample was 6.65 × 10^−3^ S·cm^−1^ at 26 °C, as shown in Figure 3b. Zhu et al. [52] prepared Li_1.5_Al_0.5_Ge_1.5_(PO_4_)_3_ glass-ceramic SSEs by the melt-quenching method, and the effects of different crystallization temperatures were investigated by XRD, SEM and NMR. SEM images are shown in Figure 3c. The results show that the formation of amorphous phases, cracks and voids can be effectively controlled by adjusting the crystallization temperature, thus improving the ion transport at the grain boundaries. Nikodimos et al. [53] prepared Sc-doped Li_1+x+y_Al_x_Sc_y_Ge_2−x−y_(PO_4_)_3_ by melt quenching and found that it has high ionic conductivity and good contact properties with the anode. Meanwhile, the material also showed an electrochemical stability window of up to 7.5 V vs. Li/Li^+^.

The melt-quenching method for the preparation of NASICON-type glass-ceramic SSEs is still the mainstream preparation method today, and some recent studies on the preparation of NASICON-type glass-ceramic SSEs by melt quenching are summarized in Table 1. However, other methods have also been used to prepare this type of glass-ceramic SSE. Yi et al. [54] prepared Li_1.7_Al_0.3_Ti_1.7_Si_0.4_P_2.6_O_12_ glass-ceramic SSEs by the liquid-feed flame spray pyrolysis (LF-FSP) process, and the ionic conductivity reached 7.7 × 10^−4^ S·cm^−1^ at room temperature. In addition, microwave sintering [55], spark plasma sintering [56] and hot-press sintering [57] methods for the preparation of NASICON-type glass-ceramic SSEs have been reported.

#### 3.1.2. Other Oxide Glass-Ceramic Systems

In addition to NASICON compounds, there are other oxides that can be used as SSEs. These oxide glass-ceramic SSEs are prepared by different methods, such as mechanochemical methods and melt-quenching methods. Mechanochemical preparation of glass-ceramic SSEs is mechanically treating the raw material to convert mechanical energy to the energy of chemical reaction [66,67,68]. Tatsumisago et al. [69] obtained 90Li_3_BO_3_·10Li_2_SO_4_ glass-ceramic SSEs with an ionic conductivity of 1.4 × 10^−5^ S·cm^−1^ by the mechanochemical method at room temperature, as shown in Figure 4a. Yoneda et al. [70] prepared 90Li_4_SiO_4_-10Li_2_SO_4_ glass-ceramic SSEs by the mechanochemical method, and then assembled ASSLIBs with Li-In/LiNi_1/3_Mn_1/3_Co_1/3_O_2_ without high-temperature sintering. The melt-quenching method also can be used to prepare oxide glass-ceramic SSEs. Widanarto et al. [71] prepared (85 − x)TeO_2_-xLi_2_O·15ZnO (x = 0, 5, 10, 15 mol%) by the melt-quenching method; SEM images are shown in Figure 4b. The study indicates that high-quality zinc-tellurite glass-ceramic SSEs with improved ionic conductivity can be obtained by proper control of temperature, AC frequency (AC) and Li_2_O concentration. Tezuka et al. [72] prepared Li_4_B_7_O_12_Cl glass-ceramic SSEs by the melt-quenching method with an ionic conductivity of 4.6 × 10^−4^ S·cm^−1^ at 200 °C and the conductivity activation energy was 0.52 eV.

In addition to the two preparation methods already presented, oxide glass-ceramic SSEs can also be prepared by other methods. Nagao et al. [73] prepared 90Li_3_BO_3_·7Li_2_SO_4_·3Li_2_CO_3_ glass-ceramic SSEs by the mechanical ball-milling method with an ionic conductivity of 1 × 10^−5^ S·cm^−1^ at room temperature. Okumura et al. [74] prepared Li_2.2_C_0.8_B_0.2_O_3_ glass-ceramic SSEs by the spark plasma sintering (SPS) process. The Li^+^ conductivity at 30 °C was 2.1 × 10^−6^ S·cm^−1^. Shin et al. [75] prepared garnet-type Li_7_La_3_Zr_2_O_12_-8wt%Li_3_BO_3_ glass-ceramic SSEs by low-temperature sintering using Li_3_BO_3_ glass-ceramic as a sintering additive with an ionic conductivity of 1.94 × 10^−5^ S·cm^−1^ at room temperature.

### 3.2. Sulfide Glass-Ceramic SSE Systems

Compared to oxide SSEs, sulfide SSEs have been intensively studied in recent years due to their advantages such as higher ionic conductivity at room temperature and cheaper raw material. Sulfides can be processed into three forms: glass, glass-ceramic and crystalline. Glass-ceramic SSEs generally have better performance than the other two forms. Therefore, the sulfide glass-ceramic SSE system, represented by the glass-ceramic SSEs in the Li_2_S-P_2_S_5_ binary system (LPS glass-ceramic SSEs), has been studied extensively in recent years.

#### 3.2.1. Li_2_S-P_2_S_5_ Binary System

The Li_2_S-P_2_S_5_ binary system has several compounds, including Li_2_P_2_S_6_, Li_4_P_2_S_6_, Li_7_P_3_S_11_ and Li_3_PS_4_, as shown in Figure 5 [18]. In the xLi_2_S-(100 − x)P_2_S_5_ (x, molar percent) system, xLi_2_S-(100 − x)P_2_S_5_ glass-ceramic SSEs containing 70% < x < 80% were the most studied, for example, 70Li_2_S-30P_2_S_5_ [76], 75Li_2_S-25P_2_S_5_ [77] and 78Li_2_S-22P_2_S_5_ [78]. Therefore, we only briefly introduce the crystal structures of Li_7_P_3_S_11_ and Li_3_PS_4_.

Li_7_P_3_S_11_ is usually obtained from 70Li_2_S-30P_2_S_5_ by heat treatment and has a very high Li^+^ conductivity with low room temperature conduction activation energy [76]. Its crystal structure has trigonal symmetry, space group P-1, with two Li_7_P_3_S_11_ units per cell [18]. The crystal structure can be regarded as consisting of PS_4_^3−^ tetrahedra and P_2_S_7_^4−^ 4-bis-tetrahedra, and Li^+^ is mainly distributed in the interstices between the tetrahedra and bis-tetrahedra [79]. Ceder et al. [80] considered that the tetrahedra composed of S^2−^ in Li_7_P_3_S_11_ are face-centered cubic-stacked, which can provide a lower conduction activation energy and facilitate the rapid transport of Li^+^.

Li_3_PS_4_ belongs to the Li_2_S-P_2_S_5_ binary system of 75Li_2_S-25P_2_S_5_, which is assembled into ASSLIBs under the same conditions and has better cycling performance than Li_7_P_3_S_11_ [81]. Li_3_PS_4_ has four main crystalline forms: β-Li_3_PS_4_, γ-Li_3_PS_4_, α-Li_3_PS_4_ and δ-Li_3_PS_4_. In 2011, Homma et al. [82] reported β-Li_3_PS_4_ by heating the γ-Li_3_PS_4_ to 300 °C. Although β-Li_3_PS_4_ did not receive much attention initially, β-Li_3_PS_4_ glass-ceramic SSEs synthesized by ball milling were later found to have high ionic conductivity. It is now commonly believed that β-Li_3_PS_4_ consists of hexagonally close-packed sulfide ions with Li and P in the generated interstitials. It is suggested that the distortion of the close-packed arrangement due to the difference in size and binding properties of Li and P is responsible for the higher ionic conductivity of β-Li_3_PS_4_ than γ-Li_3_PS_4_ [80].

#### 3.2.2. Synthesis of LPS Glass-Ceramic SSEs

Currently, most of the reported LPS glass-ceramic SSEs have been prepared mainly by mechanical ball milling. Mechanical ball treatment can be controlled by the proportion of the reagents and milling beads, milling speed and time in order to carry out the chemical process [83]. The material prepared by this process is usually in the glassy state and requires heat treatment of the parent glass to crystallize it in order to obtain the glass-ceramic SSEs. Kim et al. [84] prepared 78.3Li_2_S∙21.7P_2_S_5_ with an ionic conductivity of 6.3 × 10^−4^ S·cm^−1^ at room temperature by the mechanical ball-milling method and subsequent heat treatment. During heat treatment, it is extremely important to control the temperature and time of the heat treatment to control the crystal microstructure and, thus, improve the performance of the electrolyte. Lu et al. [85] successfully controlled the microstructure of 75Li_2_S∙25P_2_S_5_ based on the precipitation kinetics and effective medium approach and prepared the sample by mechanical ball milling. The microstructure of the prepared SSEs was well controlled, and its electrical conductivity increased by 80%. The LPS glass-ceramic SSEs prepared by this method were also used to assemble ASSLIBs with good cell performance. Yu et al. [86] prepared Li_7_P_3_S_11_ by the mechanical ball-milling method and subsequent annealing heat treatment for assembling ASSLIBs with Li_2_S/Li_7_P_3_S_11_/Li-In structure. The ASSLIBs provided a discharge specific capacity of 1139.5 mAh g^−1^ during the initial cycle and still maintained a discharge specific capacity of 850.0 mAh g^−1^ after 30 cycles. Wang et al. [87] also prepared Li_7_P_3_S_11_ by the mechanical ball-milling method as well as heat treatment and assembled Li-S cells with FeS_2_/Li_7_P_3_S_11_/Li-In structure, which provided 620.8 mAh g^−1^ initial discharge capacity at 0.1C at room temperature.

It may be supposed that the heat generated by the high-energy collision between the raw material and the grinding medium at room temperature is sufficient to partially melt and recrystallize the material. Trevey et al. [88] successfully prepared Li_2_S-GeS_2_-P_2_S_5_ glass-ceramic SSEs by the SSBM process for the assembly of ASSLIBs with a Li/Li_2_S-GeS_2_-P_2_S/LiCoO_2_ structure, which exhibited a discharge capacity at the second cycle of 129 mAh g^−1^. In addition to the mechanical ball-milling method, the melt-quenching method can also be used to prepare LPS glass-ceramic SSEs. Seino et al. [89] prepared the parent glass by melt quenching. The glass powder was compressed at 94 MPa first, and then heated at 280 °C or 300 °C for 2 h. The prepared 70Li_2_S-30P_2_S_5_ glass-ceramic sample had a very high ionic conductivity of 1.7 × 10^−2^ S·cm^−1^ at room temperature and a minimum conduction activation energy of 17 kJ·mol^−1^, as shown in Figure 6a. Preefer et al. [90] prepared Li_7_P_3_S_11_ samples by using a rapid assisted-microwave procedure, which showed good ionic conductivity at room temperature.

In addition, the liquid-phase synthesis method allows the preparation of more homogeneous electrolyte materials and also has the potential for large-scale industrial preparation [32]. Therefore, the preparation of LPS glass-ceramic SSEs by liquid-phase synthesis is a new method in recent years. The method is based on the addition of raw materials to organic solvents, followed by heat treatment to remove the organic solvents, and finally sintering the products to produce LPS glass-ceramic SSEs. Xu et al. [33] first ground the raw materials into powder, then dispersed the powder in acetonitrile (ACN) solution separately, and prepared Li_7_P_3_S_11_ samples by two-step heat treatment. The preparation process is shown in Figure 6b. At room temperature, this sample showed an ionic conductivity of 9.7 × 10^−4^ S·cm^−1^ and a low activation energy of 31.2 kJ·mol^−1^. Calpa et al. [34] prepared the Li_7_P_3_S_11_ sample by liquid-phase treatment under ultrasonic treatment, achieving a high ionic conductivity of 1.0 × 10^−3^ S·cm^−1^ at 22 °C and a low activation energy of 31.2 kJ·mol^−1^. Choi et al. [35] prepared 75Li_2_S-25P_2_S_5_ glass-ceramic SSEs using the low-temperature solution technique (LTST), which reduced the ionic conductivity of this type of material but increased the interface area between the LiCoO_2_ cathode and 75Li_2_S-25P_2_S_5_ electrolyte, thus improving the cycling performance of the battery.

#### 3.2.3. Enhancement of LPS Glass-Ceramic Performance

LPS glass-ceramic SSEs have high ionic conductivity, but most still fall short of existing organic liquid electrolytes. Meanwhile, LPS glass-ceramic SSEs are more sensitive to moisture. Once in a humid environment, they can produce toxic H_2_S gas leading to structural changes in the electrolyte as well as the decay of ionic conductivity [91]. In addition, LPS glass-ceramic SSEs also generally suffer from a narrow electrochemical window. Therefore, it is necessary to adopt some methods to enhance the various performance of LPS glass-ceramic SSEs to promote their practical application.

Currently, most of the research reports focus on the enhancement of various properties of LPS glass-ceramic SSEs by doping methods. This method is mainly used to enhance the performance of the electrolyte by creating defects in the crystal structure of the material and expanding the Li^+^ transport channels. In the reported studies, the main doped substances include oxides, sulfides, halogenated compounds and some other compounds [92,93]. In addition to single-phase doping, two-phase co-doping or even three-phase doping can be used to improve the performance of LPS. In conclusion, optimization of each property including ionic conductivity, material stability and interfacial properties with electrodes is the key to optimizing material properties by doping. Oxides including Li_2_ZrO_3_ [94], LiSO_4_ [95], Li_2_O [96], ZnO [97], LiNO_3_ [98] and Nb_2_O_5_ [99] can effectively enhance the ionic conductivity performance of SSEs materials by doping. 70Li_2_S-(30 − x)P_2_S_5_-xLi_3_PO_4_ was successfully prepared by Huang et al. [100], exhibiting 1.87 × 10^−3^ S·cm^−1^ with a minimum activation energy of 18 kJ/mol when x = 1% mol. The assembled Li-In/70Li_2_S-29P_2_S_5_-1Li_3_PO_4_/LiCoO_2_ cell exhibited a discharge specific capacity of 108 mAh g^−1^, as shown in Figure 7a. The impedance spectrum EIS analysis revealed that the doping with Li_3_PO_4_ reduced the interfacial resistance between the electrode and electrolyte, as shown in Figure 7b. Lu et al. [94] prepared 99(70Li_2_S-30P_2_S_5_)-1Li_2_ZrO_3_ glass-ceramic SSEs with a high ionic conductivity of 2.85 × 10^−3^ S·cm^−1^. After being assembled into ASSLIBs, they exhibited a higher cell cycling performance. Tsukasaki et al. [101] successfully prepared (100 − x)Li_3_PS_4_-xZnO, and found that Li_3_PS_4_ doped with 10% or 20% ZnO could better balance the performance of thermal stability, moisture resistance and ionic conductivity by DSC analysis

In addition to oxides, sulfides including GeS_2_ [102], P_2_S_3_ [103], SnS_2_ [104], Ni_3_S_2_ [105] and LiSnS_4_ [106] can also be used for the doping of LPS glass-ceramic SSEs. (100 − x)(70Li_2_S-30P_2_S_5_)-xFeS_2_ glass-ceramic SSEs were prepared by Zhou et al. [107] and then characterized by solid-state NMR. It was found that FeS_2_ doping could controllably adjust the crystalline part in the glass-ceramic SSEs to achieve excellent ionic conductivity, as shown in Figure 7c. Cells with the structure FeS_2_ composite/99.5(70Li_2_S-30P_2_S_5_)-0.5FeS_2_/Li–Ln showed higher initial capacity and better cycling performance than those with the structure FeS_2_ composite/70Li_2_S-30P_2_S_5_//Li–In. Otoyama et al. [108] added LiSnS_4_ into Li_3_PS_4_ to form the LiSnS_4_-Li_3_PS_4_ system, which improved the ionic conductivity as well as the air stability of the glass-ceramic SSEs without affecting the electrochemical stability. Halogen compounds such as LiX (X = F, Cl, Br, I) [109,110] and Li(BH_4_)_0.75_I_0.25_ [111], etc., are also widely used for doping. Tatsumisago et al. [112] systematically investigated the doping effect of LiX (X = F, Cl, Br, I) on Li_7_P_3_S_11_, and their results showed that the doping with LiBr was most effective in enhancing the ionic conductivity of Li_7_P_3_S_11_ glass-ceramic SSEs. Further study by Zhao et al. [113] showed that LiBr does not enter the lattice but exists in the interstices between the Li_7_P_3_S_11_ lattice. The high electronegativity of Br reduces the electron cloud density on the surface of P_2_S_7_^4−^ and PS_4_^3−^ units, decreasing their binding to Li^+^, and, thus, increasing the ionic conductivity.

With the in-depth study of doping methods, multiphase co-doped LPS glass-ceramic SSEs have also been reported in recent years. Zhang et al. [114] investigated Li_7_P_3_S_11_ glass-ceramic SSEs co-doped with WS_2_ and LiBr by dielectric spectroscopy, and their results showed that the doped LPS-based glass-ceramic SSEs had synergistic effects in terms of ionic conductivity and interfacial compatibility. Wang et al. [115] successfully prepared Zn-, Br- and I-substituted LPSZn_0.05_Br_0.2_I_0.8_ glass-ceramic SSEs with high ionic conductivity as well as low activation energy at room temperature, as shown in Figure 7d. Additionally, the Li^+^ conductivity can be enhanced by adding a certain amount of Li as a charge carrier to the Li_7+x_P_3_S_11_ glass-ceramic SSEs [116]. The ionic conductivity can also be improved by reducing the grain boundaries of the material through hot-press densification and adjusting and optimizing the heat treatment parameters in the material preparation method [89,117].

**Figure 7 materials-16-02655-f007:**
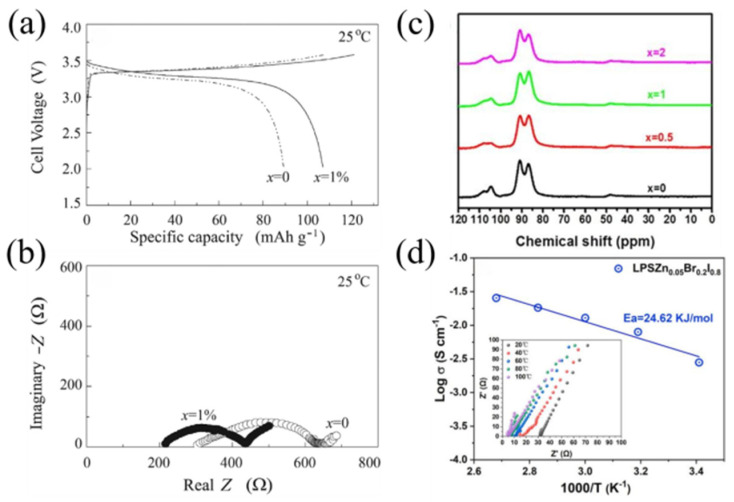
(**a**) Charge and discharge curves of Li-In/70Li_2_S-(30 − x)P_2_S_5_-xLi_3_PO_4_/LiCoO_2_ battery; (**b**) electrochemical impedance spectra of Li-In/70Li_2_S-(30 − x)P_2_S_5_-xLi_3_PO_4_/LiCoO_2_ battery. Reprinted with permission from ref. [100]. Copyright 2019 Elsevier. (**c**) The ^31^P MAS NMR spectra of (100 − x)(70Li_2_S-30P_2_S_5_)-xFeS_2_ (x = 0, 0.5, 1, 2) glass-ceramic samples. Reprinted with permission from ref. [107]. Copyright 2020 Elsevier. (**d**) Conductivity and impedance data for LPSZn_0.05_Br_0.2_I_0.8_ glass-ceramic SSEs. Reprinted with permission from ref. [115]. Copyright 2022 Elsevier.

The recent studies on LPS electrolytes are summarized, including ionic conductivity, energy density of assembled cells and electrochemical window, as shown in Table 2. The data presented in Table 2 show that doping, optimization of preparation methods and some other methods can significantly improve the properties of LPS glass-ceramic SSEs.

## 4. Interfacial Problems of Solid-State Electrolytes

Glass-ceramic SSEs have better interfacial properties than polycrystalline ceramic SSEs due to the presence of amorphous glass [20]. However, the interfacial problem between glass-ceramic SSEs and positive/negative electrodes is still an important challenge limiting the practical application of ASSLIBs [118]. Therefore, many studies on the interfacial properties of glass-ceramic SSEs with electrodes have also been reported. In this chapter, we will first briefly introduce the interfacial problem and its optimization methods, and then we will give an overview of the research on the interfacial properties of glass-ceramic SSEs.

### 4.1. Interface Problems and Optimization Methods

The interface problems between SSEs and electrodes include poor interfacial wettability and compatibility. This is manifested by a small interfacial contact area leading to insufficient contact, interfacial reactions and high interfacial resistance [119,120,121,122,123]. For ISEs, especially oxides, the interfacial problems are mainly due to high interfacial resistance caused by their rigid nature, poor electrode–electrolyte interfacial compatibility and technological difficulties [124]. For sulfide glass-ceramic SSEs, the poor stability in air is also responsible for their poor interfacial properties. This is due to the fact that sulfide glass-ceramic SSEs react with water in air to produce toxic H_2_S gas, leading to the destruction of their structure, which leads to a series of problems such as the reduction in ionic conductivity [91]. In addition, consistency of composition and structure between the grain boundaries and the bulk phase are important for guaranteeing a low Li^+^ transport resistance across the grain boundaries interface. Chemical composition and structural deviations would result in weak interactions between the framework and charge carriers, discontinuous pathways and a higher energy barrier for Li^+^ conduction. [125]. The tight contact at the interface between the electrode and the SSEs is the key factor to improve the electrochemical performance of all ASSLIBs. There are three main aspects of current studies, including electrodes, electrolytes and the transition layer introduced between electrodes and SSEs to improve the interfacial properties.

For electrodes, designing an excellent composite electrode is important to enhance the interfacial properties [126]. Wang et al. [127] designed a Li-metal negative electrode with PEO-50000 (LiTFSI) film and obtained good interface by assembling into a cell of Li-PEO-500000 (LiTFSI)/LAGP-PEO1/LiMFP, as shown in Figure 8. Zhou et al. [128] then used organic quinone cathode 5,7,12,14-pentaerythritone (PT) to prepare an ASSLIB with a glass-ceramic 70Li_2_S-30P_2_S_5_ sulfide electrolyte, which exhibited excellent rate performance and cycling performance. The reason for this is that the inherently low Young’s modulus of the PT electrode effectively prevents contact loss at the interface.

The transition layer between the electrode and the SSEs can also enhance the interfacial properties [129,130]. Kato et al. [131] found that the insertion of Au films between the Li metal and the solid electrolyte can effectively maintain stable Li dissolution and deposition, thereby improving the utilization of Li-metal electrodes in all-solid-state batteries. Liang et al. [132] then introduced a Li^+^ conduction buffer layer on the cathode surface to construct a well-matched interface between the cathode and SSEs.

In addition to the two mentioned methods, the interfacial properties between electrodes and SSEs can be enhanced by synthetic methods, modification of electrolytes and so on.

### 4.2. Enhancement of Interfacial Properties of Oxide Glass-Ceramic SSE Systems

The improvement of the interfacial properties of LATP and LAGP can be achieved in various ways, such as optimization of the preparation method, compounding with PSEs to form CSEs, introduction of thin films on the electrolyte surface and structural modifications. Structural modification of NASICON-type glass-ceramic SSEs is currently the most prominent method to enhance interfacial properties. Jadhav et al. [133] prepared LAGP glass-ceramic materials doped with B_2_O_3_, and the B_2_O_3_ can stabilize LAGP in weak acid and weak base environments. Saffirio et al. [134] prepared Li_1.4_Al_0.4_Ge_0.4_Ti_1.4_(PO_4_)_3_ doped with 0.05% B_2_O_3_ and showed that the doping with B_2_O_3_ enhanced the anodic oxidation stability of the material and reduced the grain boundary resistance. This shows that the doping with B_2_O_3_ is helpful for the interfacial properties of LAGP-type glass-ceramic SSEs. Yamamoto et al. [135] successfully prepared LASGTP by co-doping LATP with Si and Ge, and cells with the structure of LiCoO_2_/LASGTP/Pt were assembled. The crystalline phases in the LASGTP glass matrix are composed of Li_1+x_Al_x_Ge_y_Ti_2−x−y_P_3_O_12_ (main-phase), Li_1+x+3z_Al_x_(Ge,Ti)_2−x_(Si_z_PO_4_)_3_ (sub-phase) and AlPO_4_. They suggested that the insertion of Li into the LASGTP to form an amorphous phase and the gradual distribution of Li around the interface would lead to irreversible in situ formation of the anode in the LASGTP and produce low interfacial resistance.

The interfacial properties of NASICON-type glass-ceramic SSEs can also be improved by introducing thin films on the electrolyte surface. Liu et al. [136] sputtered amorphous Ge films on the LAGP surface, which not only inhibited the reduction reaction between Ge^4+^ and the Li-metal negative electrode, but also made a close contact between the Li-metal negative electrode and LAGP electrolyte. It was demonstrated by XPS characterization that the Ge film was formed only on the surface of SSEs, as shown in Figure 9a. Hu et al. [137] sputtered a metal Bi film on LAGP, which not only suppressed the unfavorable reaction between the LAGP electrolyte and Li-metal anode, but also improved their compatibility. The SEM image of the electrode–electrolyte interface cross-section is shown in Figure 9b.

In addition, by improving the preparation methods such as heat treatment conditions, the interface properties can be improved to a certain extent [138]. The formation of CSE through the composite of glass-ceramic SSEs and PSEs is also a mainstream direction to improve the interfacial properties [139].

### 4.3. Enhancement of Interfacial Properties of Sulfide Glass-Ceramic SSE Systems

For sulfide glass-ceramic SSEs, the enhancement of the interfacial properties relies mainly on the structural modification by the dopants such as Fe_2_S [107], LiBr [113], LiNO_3_ [98], LiI [109] and SeS_2_ [140]. In the previous section, we focused on the performance improvement of LPS glass-ceramic SSEs, so here we only present its improvement in interfacial properties. Feng et al. [141] successfully prepared new glass-ceramic SSEs of Li_10_P_3_S_12_I by mixing Li_2_S, P_2_S_5_ and LiI in a certain ratio through solid-phase reaction. Li_10_P_3_S_12_I has higher interfacial stability and lower interfacial resistance than thiophosphate. This is mainly because Li_10_P_3_S_12_I generates LiI at the interface of the electrode as well as the electrolyte during the electrochemical cycle, and LiI contributes to the improvement of the interfacial stability. Additionally, it has been shown that the introduction of LiI could inhibit the growth of Li dendrites in LPS glass-ceramics, thus improving the cycling performance of the cell [109]. Wu et al. [140] successfully prepared SeS_2_-doped 70Li_2_S-30P_2_S_5_, and observed the interface by EIS analysis and SEM. The result indicates that the addition of SeS_2_ contributes to the reduction of the interfacial resistance, as shown in Figure 10a–d. Through the doping of LiNO_3_, Ahmad et al. [98] obtained a thermodynamically stable Li_2_O and Li_3_N solid electrolyte interface (SEI) at the interface between the electrode and the Li-metal anode, thus inhibiting the occurrence of interfacial reactions and the growth of Li dendrites.

In addition to structural modifications, less research has been conducted to enhance the interfacial properties of LPS glass-ceramic SSEs by interfacial engineering of the coated films and optimization of the heat treatment conditions. Wei et al. [117] showed that the total interfacial resistance of Li/SE/Li cells decreased by more than an order of magnitude with increasing heat treatment annealing temperature. However, too-high annealing temperature resulted in the formation of a low conductivity phase of Li_4_P_2_S_6_ resulting in higher interfacial resistance. Xu et al. [142] assembled the LiNbO_3_@LiCoO_2_/Li_7_P_3_S_11_/Li cell using methoxyperfluorobutane (HFE)-coated/permeable Li_7_P_3_S_11_ glass-ceramic SSEs with a LiF-coated Li-metal anode, showing high reversible discharge capacity as well as cycling performance, as shown in Figure 10e,f.

## 5. Conclusions and Perspective

Lithium batteries are widely used in power and energy storage applications due to their high energy density, good cycling performance and no memory characteristics. However, the current liquid electrolyte-based LIBs in the market are approaching the upper limit of their theoretical specific capacity and the safety issues will make it difficult to meet the future power needs of electric vehicles. The ASSLIBs based on SSEs can advance the application of the Li-metal anode to make a Li battery with higher theoretical specific capacity and better safety performance. Glass-ceramic SSEs have both polycrystalline ceramic and amorphous glass phases, and, thus, have the advantages of high ionic conductivity, Li^+^ transfer number and good interfacial properties. This review summarizes the recent research reports on glass-ceramic SSEs and briefly introduces the ion transfer mechanism, preparation methods, performance enhancement and their interfacial issues with electrodes. However, the current research reveals that glass-ceramic SSEs are still challenging from the perspective of practical application.

Although the glass-ceramic SSE has a high ionic conductivity (10^−4^~10^−2^ S·cm^−1^), there is still a gap to its practical application. This is mainly because LPS electrolyte materials still have problems such as water sensitivity and a narrow electrochemical window. Optimization of preparation methods and structural modifications are important to improve the properties of glass-ceramic SSEs.In addition to the properties of the materials themselves, the industrial production of the materials is another factor that hinders their practical application. Traditional solid-state reactions, mechanical ball milling and melt quenching require much time and effort. All these ways are difficult to apply to the practical production of glass-ceramic SSEs. The liquid-phase synthesis method seems to be a potential method for industrial production. However, for the present studies, the liquid-phase synthesis method is also not ready for practical production. Therefore, more research on industrial production methods for glass-ceramic SSEs is still necessary in the future.The small interfacial contact area caused by interfacial problems leads to poor contact, insufficient interfacial reactions and high interfacial resistance, which is still the most difficult obstacle to break through to further the practical application of ASSLIBs. The design of a good electrode/electrolyte contact interface through structural modification, interface engineering and optimization of preparation methods is the main way to improve the interfacial properties.

Overall, this review summarizes recent research on glass-ceramic SSEs in terms of preparation methods, characterization means, performance enhancement and electrode/electrolyte interface properties, hoping to assist in the research and practical application of ASSLIBs. Improving the performance of glass-ceramic SSE materials, expanding their production scale and designing excellent battery internal structures to promote safer and higher energy density batteries for practical applications are still the focus of future research.

## Figures and Tables

**Figure 2 materials-16-02655-f002:**
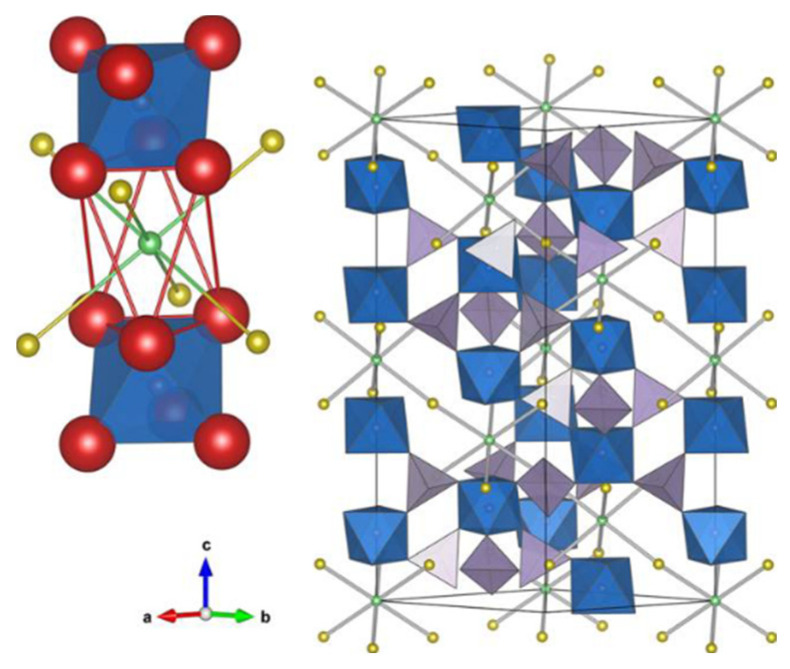
Representations of a typical NASICON structure. Blue octahedra are MO_6_ units, purple tetrahedra are PO_4_ units, green spheres are M1 sites and yellow spheres are M2 sites. Reprinted with permission from ref. [41]. Copyright 2014 American Chemical Society.

**Figure 3 materials-16-02655-f003:**
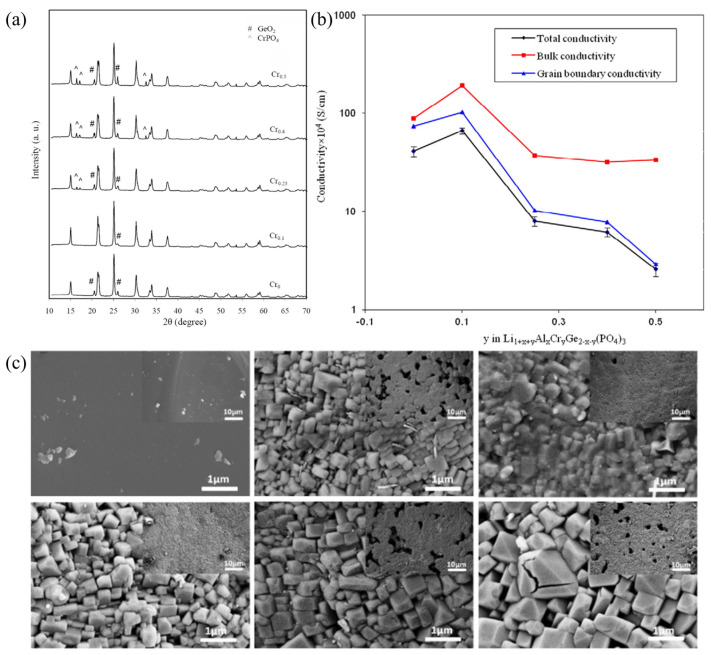
(**a**) XRD patterns of samples with different Cr contents crystallized at 850 °C for 8 h; (**b**) The total, bulk and grain boundary conductivities measured at 26 °C for Li_1+x+y_Al_x_Cr_y_Ge_2−x−y_(PO_4_)_3_ samples with different Cr contents. Reprinted with permission from ref. [51]. Copyright 2016 Elsevier. (**c**) SEM micrographs of LAGP samples in glass phase and crystallized at 750 °C, 775 °C, 800 °C, 825 °C, 850 °C. Reprinted with permission from ref. [52]. Copyright 2015 Elsevier.

**Figure 4 materials-16-02655-f004:**
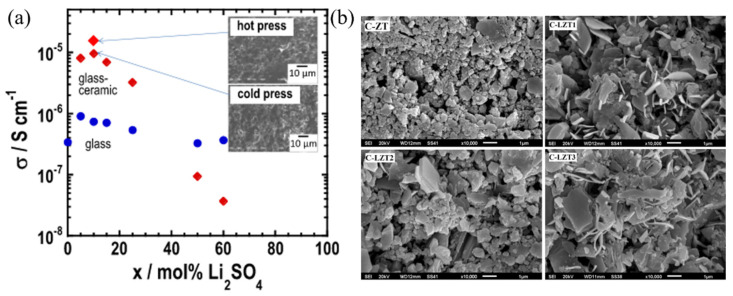
(**a**) Composition dependence of the ionic conductivity at room temperature of mechanochemically prepared Li_3_BO_3_·Li_2_SO_4_ glasses and the corresponding glass precursors heat-treated at temperatures just above the first crystallization peak. The inset shows SEM photographs of compressed particles of Li_2.9_B_0.9_S_0.1_O_3.1_ powder prepared by cold pressing at room temperature and hot pressing at 255 °C. Reprinted with permission from ref. [69]. Copyright 2014 Elsevier. (**b**) SEM images of the prepared (85 − x)TeO_2−_xLi_2_O·15ZnO (x = 0, 5, 10, 15 mol%) glass-ceramic electrolytes. Reprinted with permission from ref. [71]. Copyright 2017 Elsevier.

**Figure 5 materials-16-02655-f005:**
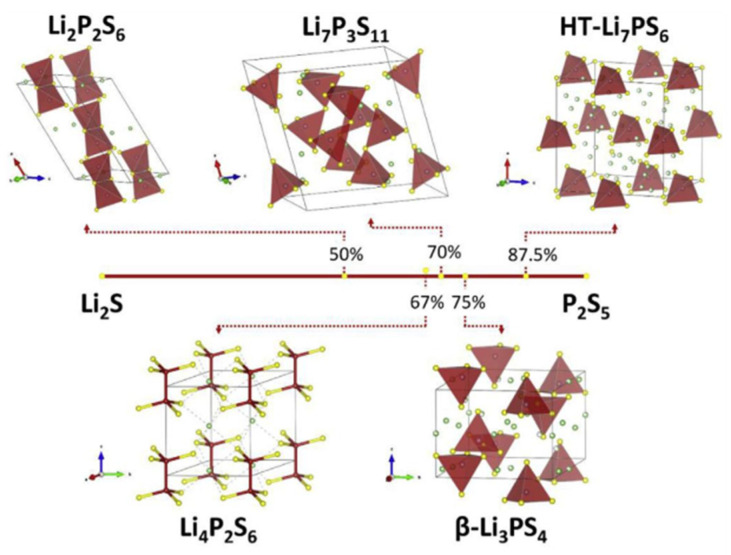
Some of the crystal structures observed in materials, formed within Li_2_S-P_2_S binary system. Reprinted with permission from ref. [18]. Copyright 2018 Elsevier.

**Figure 6 materials-16-02655-f006:**
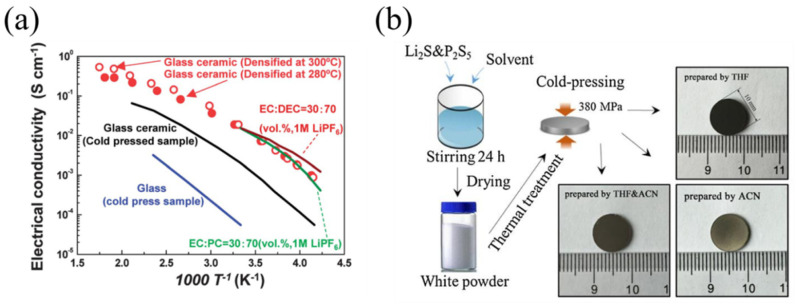
(**a**) Plots of ionic conductivity versus temperature for 70Li_2_S-30P_2_S_5_, cold-pressed glass, glass-ceramic powder and some common liquid electrolytes prepared by melt-quenching method. Reprinted with permission from ref. [89]. Copyright 2014 Royal Society of Chemistry. (**b**) Preparation process of Li_7_P_3_S_11_ samples by liquid phase synthesis. Reprinted with permission from ref. [33]. Copyright 2016 Elsevier.

**Figure 8 materials-16-02655-f008:**
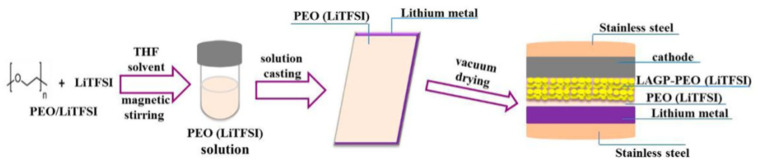
All-solid-state Li-PEO (LiTFSI)/LAGP-PEO (LiTFSI)/LiMFP cells. Reprinted with permission from ref. [127]. Copyright 2017 American Chemical Society.

**Figure 9 materials-16-02655-f009:**
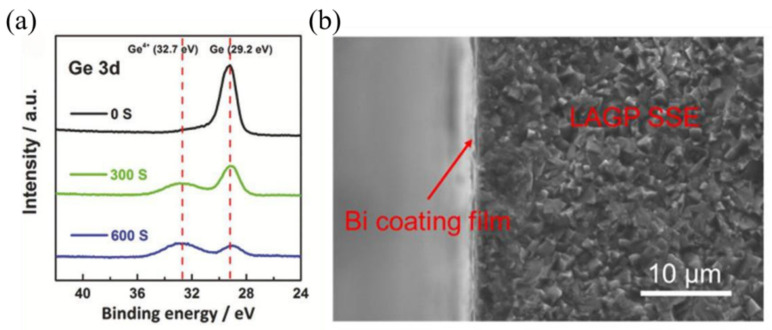
(**a**) XPS of the Ge film on LAGP pellet. Reprinted with permission from ref. [136]. Copyright 2018 WILEY-VCH Verlag GmbH & Co. KGaA, Weinheim. (**b**) Cross-sectional SEM image of the LAGP pellet coated with Bi buffer. Reprinted with permission from ref. [137]. Copyright 2020 American Chemical Society.

**Figure 10 materials-16-02655-f010:**
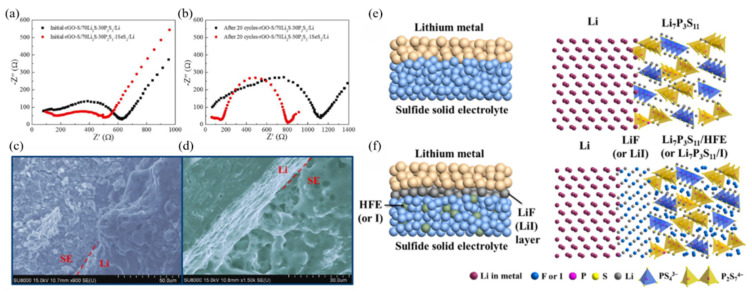
Nyquist plots of rGO/70Li_2_S-30P_2_S_5_/Li and rGO-S/70Li_2_S-29P_2_S_5_-1SeS_2_/Li ASSLIBs at 30 °C. Measurements were conducted (**a**) before and (**b**) after 20 cycles at 0.1 mA·cm^−2^; SEM images of the interface between the (**c**) 70Li_2_S-30P_2_S_5_, (**d**) 70Li_2_S-29P_2_S_5_-1SeS_2_ and Li in rGO-S/solid electrolyte/Li batteries at 0.1 mA·cm^−2^ for 100 charge–discharge cycles. Reprinted with permission from ref. [140]. Copyright 2019 Elsevier. Schematic diagrams of (**e**) Li/Li_7_P_3_S_11_ interface of ASSLIBs and (**f**) modified interface with a uniform thin LiF (or LiI) interphase layer and HFE (or I solution) infiltrated sulfide electrolyte. Reprinted with permission from ref. [142]. Copyright 2018 Elsevier.

**Table 1 materials-16-02655-t001:** Review of various parameters of NASICON-type glass-ceramic materials prepared by melt-quenching method in recent years.

Composition	Tg (°C)	Tc (°C)	Crystallization	σ (S·cm^−1^)	E_a_ (eV)	Reference
Li_1.3_Al_0.3_Ti_1.7_(PO_4_)_3_	624	660	1000 °C/0.33 h	1.3 × 10^−3^	0.27	[43]
Li_1.3_Al_0.3_Ti_1.7_(PO_4_)_3_	640	670	950 °C/70 h	1.23 × 10^−4^	0.37	[58]
Li_1.3_Al_0.3_Ti_1.7_(PO_4_)_3_-50P_2_O_5_	632	750	850 °C/10 h	8.5 × 10^−4^	0.26	[59]
Li_1.4_Al_0.4_Ge_1.6_(PO_4_)_3_	534	614	650 °C/96 h	3.8 × 10^−5^	0.52	[60]
Li_1.5_Al_0.5_Ge_1.5_(PO_4_)_3_	508.4	598.4	820 °C/2 h	5.03 × 10^−4^	0.36	[44]
Li_1.5_Al_0.5_Ge_1.5_(PO_4_)_3_	524	589	800 °C/8 h	2.9 × 10^−3^	0.29	[52]
Li_1.25_Al_0.25_Sn_0.25_Ge_1.75_(PO_4_)_3_	518	622	628 °C/1 h	3.9 × 10^−5^	0.36	[61]
Li_1.5_Al_0.33_Sc_0.17_Ge_1.5_(PO_4_)_3_			800 °C/8 h	5.8 × 10^−3^	0.28	[53]
Li_1.5_Al_0.5_Ge_1.5_(PO_4_)_3_ + 0.05Li_2_O	532	629	829 °C/6 h	7.3 × 10^−4^	0.38	[62]
Li_1.5_Al_0.5_Ge_1.5_(PO_4_)_3_-0.05B_2_O_3_	526.0	636.4	820 °C/2 h	5.5 × 10^−4^		[63]
Li_1.4_Cr_0.4_Ge_0.64_Ti_0.96_(PO_4_)_3_	623	692	900 °C/12 h	6.6 × 10^−5^	0.40	[64]
Li_1.6_Cr_0.6_Ge_0.28_Ti_1.12_(PO_4_)_3_	682.5	725.8	900 °C/2 h	2.9 × 10^−4^	0.26	[65]

**Table 2 materials-16-02655-t002:** Review of the various properties of LPS glass-ceramic SSE in recent years.

**Composition**	**σ (S·cm^−1^)**	**Structure of the Battery**	**Initial Energy Density**	**Electrochemical Window**	**Ref**
70Li_2_S∙30P_2_S_5_	1.7 × 10^−2^			−0.1~5 V vs. Li/Li^+^	[89]
Li_7_P_3_S_11_	6.3 × 10^−4^	Li_2_S/Li_7_P_3_S_11_/Li-In	1139.5 mAh/g at 0.064 mA/cm^2^		[86]
Li_7_P_3_S_11_	1.27 × 10^−3^	FeS_2_/Li_7_P_3_S_11_/Li-In	620.8 mAh/g at 0.1C		[87]
Li_7_P_3_S_11_	9.7 × 10^−4^			−0.5~5 V vs. Li/Li^+^	[33]
Li_7_P_3_S_11_	1.0 × 10^−3^			−0.5~5 V vs. Li/Li^+^	[34]
Li_7.25_P_3_S_11_	2.5 × 10^−3^	LiNi_0.8_Co_0.15_Al_0.05_O_2_/ Li_7.25_P_3_S_11_/In	106.2 mAh/g at 0.1C	2.0~3.6 V vs. Li-In	[116]
99(70Li_2_S∙30P_2_S_5_)-1Li_2_ZrO_3_	2.85 × 10^−3^	LiCoO_2_/99(70Li_2_S∙30P_2_S_5_)- 1Li_2_ZrO_3_/Li-In	134.5 mAh/g at 0.1C		[94]
Li_7_P_2.88_Nb_0.12_S_10.7_O_0.3_	3.59 × 10^−3^	Li_2_S/Li_7_P_2.88_Nb_0.12_S_10.7_O_0.3_/Li	642.1 mAh/g at 0.1C		[99]
70Li_2_S∙29P_2_S_5_-1Li_3_PO_4_	1.87 × 10^−3^	LiCoO_2_/ 70Li_2_S∙29P_2_S_5_-1Li_3_PO_4_/Li-In	108 mAh/g at 0.1C		[100]
99.5(70Li_2_S∙30P_2_S_5_)-0.5FeS_2_	2.22 × 10^−3^	FeS_2_ composite/ 99.5(70Li_2_S-30P_2_S_5_)-0.5FeS_2_/Li–Ln	543 mAh/g at 0.03 mA/cm^2^	−0.5~5 V vs. Li/Li^+^	[107]
80Li_7_P_3_S_11_-20LiBr	3.39 × 10^−3^	LiCoO_2_/80Li_7_P_3_S_11_-20LiBr/Li	120 mAh/g at 0.1 mA/cm^2^		[113]
90(0.7Li_2_S-0.29P_2_S_5_-0.01WS_2_)-10LiBr		LiCoO_2_/90(0.7Li_2_S-0.29P_2_S_5_-0.01WS_2_)-10LiBr/Li-In	129.6 mAh/g at 0.1C		[114]
75Li_2_S∙25P_2_S_5_	3.1 × 10^−4^	LiCoO_2_/75Li_2_S∙25P_2_S_5_/electrical conductive carbon	115 mAh/g at 0.05C	−1~5 V vs. Li/Li^+^	[35]
Li_3.06_P_0.98_Zn_0.02_S_3.98_O_0.02_	1.12 × 10^−3^	LiCoO_2_/LGPS/Li_3.06_P_0.98_Zn_0.02_S_3.98_O_0.02_/Li	139.1 mAh/g at 0.1C	−0.5~6 V vs. Li/Li^+^	[97]
Li_2.96_P_0.98_S_3.92_O_0.06_-Li_3_N	1.58 × 10^−3^	LiNbO_3_@NCA/ Li_2.96_P_0.98_S_3.92_O_0.06_-Li_3_N/Li	107.89 mAh/g at 0.064 mA/cm^2^	−0.5~5 V vs. Li/Li^+^	[98]
(Li_2_S)_9_-(P_2_S_5_)_3_-(Ni_3_S_2_)_1_ (LPN 9:3:1)	2.0 × 10^−3^	LPN(9:3:1)-NCM/ LPN(9:3:1)/In	117 mAh/g at 0.1C	−0.5~10 V vs. Li/Li^+^	[105]
2.5Li_3_PS_4_-0.5Li_4_SnS_4_	2.1 × 10^−3^	LiCoO_2_/2.5Li_3_PS_4_-0.5Li_4_SnS_4_/Li	93 mAh/g at 0.1C	−0.1~5 V vs. Li/Li^+^	[106]
Li(BH_4_)_0.75_I_0.25_- (Li_2_S)_0.75_∙(P_2_S_5_)_0.25_	1 × 10^−3^	TiS_2_/Li(BH_4_)_0.75_I_0.25_- (Li_2_S)_0.75_∙(P_2_S_5_)_0.25_/Li	239 mAh/g at 0.05C	−0.5~5 V vs. Li/Li^+^	[111]
78.3Li_2_S·21.7P_2_S_5_	6.3 × 10^−4^			−0.3~5 V vs. Li/Li^+^	[84]
Li_7.05_Zn_0.05_P_1.95_S_8_Br_0.2_I_0.8_	3.98 × 10^−3^			−0.5~5 V vs. Li/Li^+^	[115]

## Data Availability

Not applicable.

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
