# Peer review of "Progress and Perspective of Glass-Ceramic Solid-State Electrolytes for Lithium Batteries"

_materials, 2023, doi:10.3390/ma16072655_

Round 1
Reviewer 1 Report
The paper is review articel that summarizes the recent research on glass-ceramic electrolytes and their performance. It also briefly introduces the ion transfer mechanism, preparation methods and the interfacial issues with electrodes. This review could be of interest for the researcher starting in the field giving some practical application ideas. The paper is well written and I reccomed it for publication. Only the Figure 3 should be improved by putting text with much larger font size to become visible.
Author Response
General comment: The paper is review article that summarizes the recent research on glass-ceramic electrolytes and their performance. It also briefly introduces the ion transfer mechanism, preparation methods and the interfacial issues with electrodes. This review could be of interest for the researcher starting in the field giving some practical application ideas. The paper is well written and I recommend it for publication. Only the Figure 3 should be improved by putting text with much larger font size to become visible.
Response: we appreciate the reviewer for reviewing our manuscript. Now we have adjusted the size and arrangement of Figure 3 to ensure it can be visible.
Reviewer 2 Report
See the file attached

Reviewer 3 Report
In the research article titled “Progress and Perspective of Glass-Ceramic Solid-State Electrolytes for Lithium Batteries”, presented by Lin et al. Authors have presented the good review analysis, but there are few issues which I think should be addressed. The highlighted issues are as follow;
1. In the review I noticed the focus remained on the fabrication techniques or some how on the morphologies of the materials. But I suggest there should be discussion on the stability of the materials for the application point of view as well.
2. Importantly, authors most of the time neglect the important parameters for the performance of Li-ion batteries, their energy density and power density and working potential window.
3. Why the doped or added transition metal oxides are ignored for this study.
These are my suggestions which I think can improve the readership for the article. But as a whole good work by authors.
Round 2
Reviewer 2 Report
The authors have drastically corrected the text and in the present form it may be published n the journal